# Video Decomposition Prior: Editing Videos Layer by Layer

**Gaurav Shrivastava**
University of Maryland, College Park
gauravsh@umd.edu

**Ser-Nam Lim**
University of Central Florida
sernam@ucf.edu

**Abhinav Shrivastava**
University of Maryland, College Park
abhinav@cs.umd.edu

## Abstract

In the evolving landscape of video editing methodologies, a majority of deep learning techniques are often reliant on extensive datasets of observed input and ground truth sequence pairs for optimal performance. Such reliance often falters when acquiring data becomes challenging, especially in tasks like video dehazing and relighting, where replicating identical motions and camera angles in both corrupted and ground truth sequences is complicated. Moreover, these conventional methodologies perform best when the test distribution closely mirrors the training distribution. Recognizing these challenges, this paper introduces a novel video decomposition prior 'VDP' framework which derives inspiration from professional video editing practices. Our methodology does not mandate task-specific external data corpus collection, instead pivots to utilizing the motion and appearance of the input video. VDP framework decomposes a video sequence into a set of multiple RGB layers and associated opacity levels. These set of layers are then manipulated individually to obtain the desired results. We addresses tasks such as video object segmentation, dehazing, and relighting. Moreover, we introduce a novel logarithmic video decomposition formulation for video relighting tasks, setting a new benchmark over the existing methodologies. We evaluate our approach on standard video datasets like DAVIS, REVIDE, & SDSD and show qualitative results on a diverse array of internet videos.[1]

## 1 Introduction

Deep learning techniques, prominently convolutional networks (Krizhevsky et al., 2012) and vision transformers (Dosovitskiy et al., 2020), have emerged as the preferred choice for video-related tasks (Shrivastava & Shrivastava, 2024; Bodla et al., 2021; Shrivastava & Shrivastava, 2021; Saini et al., 2022; Shrivastava, 2021) due to their superior results over the classical techniques. Typically, these frameworks are trained using a standardized method: curate a dataset of input/corrupted and ground truth/clean sequences, design the model, and minimize the L2 loss between predictions and actual values. While this supervised approach sets state-of-the-art benchmarks, it grapples with two significant challenges. First, collecting extensive datasets with annotated pairs for every task is impractical and expensive. Second, as demonstrated by prior works like Shrivastava et al. (2023), the performance of pretrained models deteriorates when the test data slightly deviates from the training data.

To circumvent these challenges, emerging video editing methods (Lu et al., 2020a;b; 2021; Kasten et al., 2021; Shrivastava et al., 2023) were developed and they offer three key advantages: they operate directly on test sequences (also known as inference-time optimization), thereby obviating the need for collecting input and ground truth pairs; they are immune to dataset biases and generalize well to unseen examples (Shrivastava et al., 2023); and they are resource-efficient as they do not have to optimize over an external training corpus instead they optimize only on the given test input video.

---

[1]Navigate to the webpage for video results.

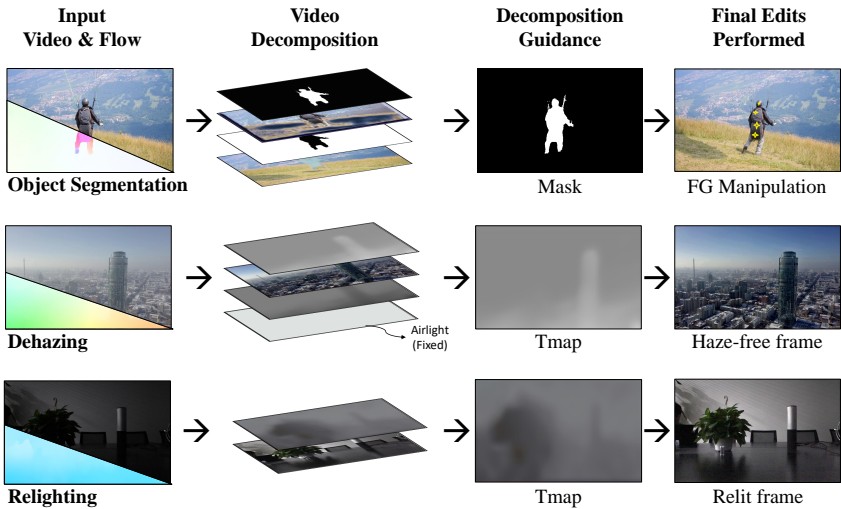

Figure 1: **Visual representation of video edits obtained using VDP**. The first row demonstrates a foreground manipulation example that leverages an object mask as the decomposition guide. This object mask is obtained by performing Unsupervised Video Object Segmentation (UVOS), which is a downstream task of video decomposition and is achieved using our proposed framework. Our approach effectively separates the foreground objects and background in the video sequence, enabling us to perform object manipulation. The second row shows the result of our approach for video dehazing, where our method effectively removes the haze from the scene. Finally, the third row showcases the effectiveness of our approach for video relighting, where our method effectively changes the lighting of the scene. Our proposed approach outperforms the state-of-the-art methods for the latter two tasks, highlighting the efficacy of our framework for video decomposition.

However, these techniques are not without limitations. They are often designed for very specific tasks, and they often perform inferior to their data-trained counterpart.

In this paper, we propose a inference-time optimization framework VDP, that does not rely on a collection of task-specific video input and ground truth sequence pairs. We took inspiration from professional video editing techniques – where sequences are broken down into multiple RGB layers. Each RGB layer is associated with a distinct opacity layer, and by manipulating these RGB layers and their associated opacity levels, desired outcomes are achieved. In this work, we emphasize that proper decomposition formulation of a video leads to visually realistic solutions for the tasks. Fig. 1 illustrates our approach for the 3 different tasks discussed. Our framework relies only on the input video and its derived optical flow to solve a wide variety of video-specific tasks.

From a technical standpoint, VDP consists of two integral modules: 'RGB-Net' that operates on the appearance aspect of the input video and '$\alpha$-Net' that operates on the associated optical flow RGB of the input video. Both of these modules are designed using the convolutional U-Net architecture. Based on the tasks, we use the appropriate video decomposition formulation and appropriate regularization terms. We then jointly optimize these convolutional U-Nets for the same.

Our work makes three main contributions. 1) To the best of our knowledge we are the first approach to utilize FlowRGBs in an inference-time optimization technique to exploit the motion cues from a query video. 2) We introduce a new formulation of the logarithmic decomposition of a video to perform a relighting task. This new formulation leads to significant improvement in performance when compared to the current baseline methods. 3) We achieve state-of-the-art performances on relevant downstream applications of video decomposition, such as video dehazing and relighting. VDP outperforms the existing inference-time optimization techniques for video object segmentation.

## 2 RELATED WORKS

Traditionally, priors played a pivotal role in transforming an 'ill-posed problem'(one with multiple plausible solutions) to a well-posed problem. Past proposed priors (Wang & Adelson, 1994; Shade et al., 1998; Shechtman et al., 2002; 2005; He et al., 2010) were very limited in their capability of learning a good representation of videos to generalize well on various scenes. All of this changed with the recent developments in deep learning techniques that considerably improve the representation of videos and deliver high-performance gains on video editing tasks.

Established works such as (Assaf Shocher, 2018; Ulyanov et al., 2018; Bell-Kligler et al., 2019; Pollak Zuckerman et al., 2020; Shrivastava et al., 2023) are test time optimization techniques, i.e., they directly work on the corrupted input image/video sequences and do not require any ground truth for training. These approaches have highlighted the importance of formulating a loss function, combined with the optimization of neural network parameters, to achieve significant results for both image and video restoration problems. However, these insights predominantly catered to limited vision tasks such as denoising, super-resolution, and inpainting. Using our approach, we are able to address video relighting, dehazing, and unsupervised video object segmentation tasks. We demonstrate this breadth of applications by leveraging the compositionality properties inherent in videos.

DoubleDIP (Gandelsman et al., 2019) employs a similar strategy, but their convolutional framework relies heavily on the appearance of the video when optimizing a task objective. This becomes a limitation when working with videos, as motion plays a critical role alongside appearance. Our method tackles this bottleneck by harnessing optical flow as a motion cue. This inclusion of optical flow not only captures motion effectively but also permits warping operations vital for downstream tasks such as edit propagation, an ability absent in DoubleDIP's decomposition framework.

Other inference-time optimization methodologies, such as (Kasten et al., 2021; Lu et al., 2021; 2020a; Bar-Tal et al., 2022) are very task-specific and utilize a singular form of video composition formulation, i.e., $\alpha$-blending. Techniques like these are heavily reliant on user-generated masks or image-centric baselines. In contrast to these methods, our proposed method can tackle a wide variety of tasks. In addition, our framework does not rely on pretrained image segmentation networks, thereby enhancing its robustness, especially when confronted with unfamiliar objects in videos. Method (Ye et al., 2022) is another inference-time optimization technique that avoids the use of pretrained image segmentation but encounters issues with objects whose appearances fluctuate over time, perhaps due to shifts in lighting. Such a limitation arises from their approach of encapsulating an object's global appearance in videos into a single canonical texture layer. In contrast, our proposed methodology takes advantage of frame-by-frame appearance and motion cues, making our model significantly more resilient to variations in lighting.

# 3 VIDEO DECOMPOSITION PRIOR FRAMEWORK

Given a video sequence denoted as $\{X_t\}_1^T$, this section delves into our framework's methodology to decompose the video into multiple layers. To facilitate this decomposition, our framework requires an associated forward optical flow of the input video, represented as $\{F_{t \to t+1}\}_2^T$. The method for obtaining this optical flow is elaborated upon in the supplementary material (Supp. Sec. H). Subsequently, we will detail the design of the modules incorporated within VDP and outline the generic structure of our loss function.

## 3.1 MODULES OVERVIEW

VDP framework consists of the following two modules.

**(I) RGB layer predictor module (RGBnet):** RGBnet (functional notation $f_{\text{RGB}}$) is a fully convolutional bottleneck network (Ronneberger et al., 2015). Given that we only optimize the weights over a single video, a shallow convolutional U-Net is sufficient for the task. RGBnet takes image frame $X_t$ of the video sequence as input and outputs a RGB layer (a 3 channel layer of same resolution as the input frame $X_t$).

**(II) $\alpha$ layer predictor module ($\alpha$-net):** $\alpha$-net (functional notation $f_\alpha$) architecture is similar to RGBnet, i.e., a fully convolutional bottleneck network. We utilize a shallow convolutional U-Net for predicting the Tmaps or opacity layer. $\alpha$-net takes flow-RGB (RGB representation of forward optical flow denoted as $F_{t \to t+1}^{\text{RGB}}$) feature maps of the video sequence as input and outputs the Tmap/opacity layer (a 1 channel layer of same resolution as the input frame $X_t$).

More details about the convolutional U-Net architecture is discussed in Supp. Sec. A.

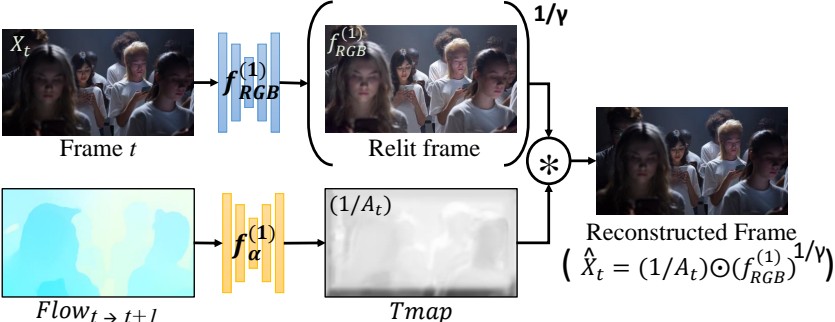

Figure 2: **VDP for relighting a video sequence.** In this pipeline, the input video frame $t$ is fed into a shallow U-Net denoted by $f_{\text{RGB}}^{(1)}$. While the flow-RGB is given as input to a separate shallow U-Net network denoted by $f_\alpha^{(1)}$. **The intermediate output of $f_{\text{RGB}}^{(1)}$ is a re-lit version of the input frame** $t$. While the output of $f_\alpha^{(1)}$ is transmission maps or Tmap denoted by $1/A_t$. It is important to note that $\gamma^{-1}$ is also treated as a trainable parameter in the pipeline. After obtaining the reconstructed frame $t$, we apply reconstruction and warping losses respectively to optimize $f_{\text{RGB}}^{(1)}, f_\alpha^{(1)}, \gamma^{-1}$.

### 3.2 VDP: OPTIMIZATION OBJECTIVE FOR GENERIC VIDEO DECOMPOSITION

VDP framework decomposes the input video into multiple layers and then composite these layers again to reconstructs the same video input. In this process, the **intermediate outputs** generated by VDP are of particular interest. In this section, we will discuss a generic structure of the loss function that will help ensure temporal consistency and faithful reconstruction of the original input video.

**Loss Objective:** The loss objective used to optimize our framework for different tasks is given by the following Eqn 1,

$$\mathcal{L}_{\text{final}} = \lambda_{\text{rec}}\mathcal{L}_{\text{rec}} + \lambda_{\text{warp}}\mathcal{L}_{\text{warp}}. \tag{1}$$

**Reconstruction Loss:** We use a combination of L1 and perceptual loss (Zhang et al., 2018) to facilitate faithful reconstruction of the original input video, which is given by,

$$\mathcal{L}_{\text{rec}} = \sum_t \|X_t - \hat{X}_t\|_1 + \|\phi(X_t) - \phi(\hat{X}_t)\|_1. \tag{2}$$

where, $\phi(\cdot)$ denotes embeddings obtained from the pre-trained VGG network trained on Imagenet (Deng et al., 2009).

**Optical Flow Warp Loss:** The intermediate output of interest $X_t^o$, should follow the same temporal coherence as the input video $X_t$ at any given timestamp. This is ensured by the optical-flow warp loss,

$$\mathcal{L}_{\text{warp}} = \sum_t \|F_{t-1 \to t}(X_{t-1}^o) - X_t^o\| \tag{3}$$

where, $F_{t-1 \to t}$ denotes the forward optical flow between the frames $X_{t-1}$ and $X_t$. Refer to Supp. Sec. I for details about the optical-flow based operations.

The subsequent sections will discuss various applications of our VDP framework.

## 4 VIDEO RELIGHTING

During the nighttime, less amount of luminescence leads to darker capture of a scene. Traditionally, gamma correction is used to relight such captured scene as gamma is what translates well between our eye's light sensitivity and that of the camera. Mathematically, gamma correction can be defined as given by the Eqn. 4

$$X_t^{\text{out}} = A_t \odot (X_t^{\text{in}})^\gamma \quad \forall t \in (1, T], \tag{4}$$

where, $X_t^{\text{out}}$ is the output of gamma correction at time step $t$. $X_t^{\text{in}}$ is the input to which the gamma correction is applied. $A_t$ and $\gamma$ are the hyperparameters of gamma correction. An image frame is

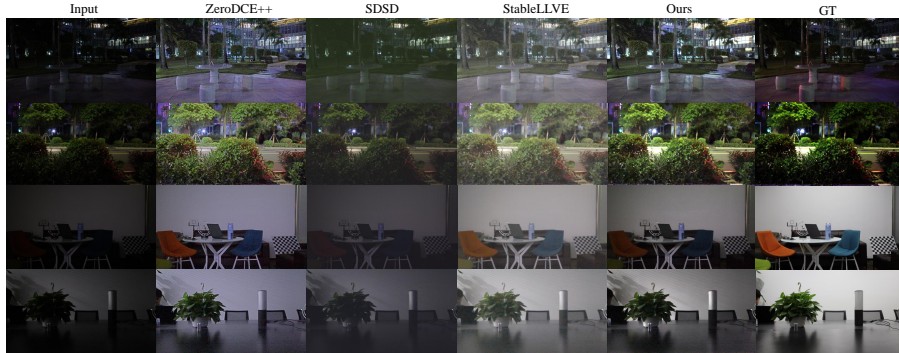

Figure 3: **Qualitative evaluation on Video Relighting benchmark:** We compare the re-lit result of our method with the baselines on the SDSD (52) dataset. We compare our method against both the image and video baselines. ZeroDCE++ (24) method is a image-based baseline while SDSD (52) and Stablellve (59) are video baselines. Please note that we have utilized the pretrained models of all the baselines to obtain qualitative results. Methods SDSD and Stablellve require training on the external dataset, while our approach operates directly on the low-light video sequence.

Table 1: **Quantitative evaluation on Video Relighting benchmarks:** For the evaluation of our method, we calculate the Avg PSNR and SSIM scores between the re-lit video and ground truth video. We use the pretrain model for the baselines that are publicly made available for calculating the PSNR and SSIM scores. For all baselines and our method, the score is calculated on the SDSD dataset (52). Also, blue (**bold**) denotes the best score.

| Metrics | ZeroDCE++ (24) | SDSD (52) | StableLLVE (59) | Ours |
|---------|----------------|-----------|-----------------|------|
| PSNR    | 26.46          | 24.92     | 26.01           | **27.92** |
| SSIM    | 0.85           | 0.73      | 0.84            | **0.90** |

either darkened or lightened up based on the value of $\gamma$ in Eqn. 4. For our model, we change the Eqn. 4 as follows,

$$\log(X_t^{\text{in}}) = \frac{1}{\gamma} \left( \log(1 \oslash A_t) + log(X_t^{\text{out}}) \right) \quad \forall t \in (1, T]. \tag{5}$$

Where $\oslash$ operation denotes element-wise division.

The goal of the video relighting task is to recover a video $\{(X_t^{\text{out}})\}_1^T$ with good lighting conditions from its poorly lit counter-part video $\{X_t^{\text{in}}\}_1^T$. We treat the relighting problem as a video decomposition problem, where one layer is the well-lit video. In addition, we model the transmission maps $1 \oslash A_t$ with our $\alpha$-net model. Further, we model $\gamma^{-1}$ as a trainable parameter that can take a value between (0,1). In summary, to find the well-lit video from the given input video, we initialize one RGBnet, one $\alpha$-net, one trainable parameter $\gamma^{-1}$, and use the Eqn. 5 for reconstruction loss. This whole pipeline is given by Fig. 2. One advantage of utilizing the convolution nets to model RGBnet($F_{RGB}$) is that convolutional nets provide a high impedance to the noise (Ulyanov et al., 2018) when reconstructing a natural image. This property proves beneficial as merely applying gamma correction to a dark image can introduce noise into the relit image.

Our final loss objective to perform the relighting task is given by Eqn. 1.

We evaluate our method for the video relighting task on the SDSD dataset (Wang et al., 2021) and compare our model against the existing state-of-the-art baselines. It is important to note that the SDSD dataset consists of low-light captured RGB videos as the input. We follow the same evaluation strategy for this dataset as the paper (Wang et al., 2021). We choose the ZeroDCE++ (Li et al., 2021), SDSD(Wang et al., 2021), and StableLLVE(Zhang et al., 2021a) approaches as baselines. SDSD and StableLLVE approaches are trained using the RGB video dataset; hence, these methods act as perfect baselines.

A qualitative evaluation can be seen in Fig. 3. Note that ours and the ZeroDCE++ approach do not require any training data. Our approach is optimized directly on the inference-time video sequence instead of other approaches that have to be trained using external data. Table. 1 represents the quantitative evaluation of our approach on the SDSD dataset. It can be observed from the table that our method outperforms the existing baselines by a significant margin in the video relighting task.

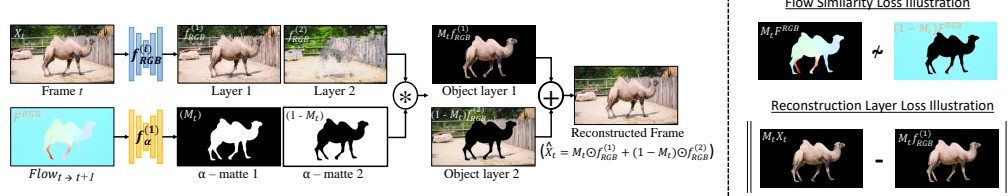

Figure 4: **Our Framework:** In this figure, we present the pipeline for the task of decomposing the video into two different components. For the above configuration, the input video frame $t$ is fed into two different shallow U-Nets denoted by $f^{(1)}_{\text{RGB}}$ and $f^{(2)}_{\text{RGB}}$ respectively. We obtain two different intermediate components of the input frame $t$ in the form of layer 1 and layer 2. Additionally, we process forward flow-RGBs using a separate shallow U-Net network denoted by $f^{(1)}_{\alpha}$. This network outputs $\alpha$ compositing maps, which are then used for blending both layers respectively to reconstruct the original input frame $t$. After the compositing, we apply various losses (reconstruction loss and regularization loss) depending on the task at hand. **(Right) Illustration of Flow Similarity loss:** we define this loss over flow-RGB image ($F^{RGB}$). This loss ensures that pixels with similar motions are grouped together. **Illustration of Reconstruction layer loss:** We define this loss over the masked object layers and masked input frame $X_t$. This loss ensures that the masked object layer such as $M_t f^{(1)}_{\text{RGB}}$ has the same appearance as the masked input frame $M_t X_t$.

## 5 Unsupervised Video Object Segmentation

Given a video sequence, the aim of unsupervised video object segmentation (UVOS) is to coherently track and segment the most salient object in a video without any human annotations. Several past works such as these (5; 7; 8; 13; 20; 29; 30; 34; 44) have made attempts to segment the primary object in a video without the use of any annotations. Unlike these works, our goal is not to detect semantic object categories but to find a decomposition that explains the scene motion and its appearance.

We start by leveraging $\alpha$-blending equation to write the reconstruction of the input video,

$$X_t = \sum_{i=1}^{L} M_t^i \odot f_{\text{RGB}}^i(X_t) \quad \forall t \in (1, T].$$ (6)

Here, $M_t^i$ denotes the alpha map for the $i^{\text{th}}$ object layer and $t^{\text{th}}$ frame. Further, $M_t^i$ is obtained using the following,

$$M_t^i = f_\alpha^i(F_{t-1 \to t}^{\text{RGB}}) \quad \forall t \in (1, n],$$ (7)

where, $F_{t-1 \to t}^{\text{RGB}}$ is the RGB image of flow estimate from frame $t-1$ to $t$. Additionally, the following condition about the alpha maps should also satisfy

$$J_{h,w} = \sum_{i=1}^{L} M_t^i,$$ (8)

where, $J_{h,w}$ denotes the all-ones matrix of size $[h, w]$.

Since the problem of video decomposition is severely ill-posed, the loss objective given by Eqn. 1 alone is not enough to find a visually plausible layered decomposition of an input video. Additionally, to find the right decomposition we need the opacity layers to be binary masks. Hence, we introduce a few other regularization losses to mitigate such challenges.

**Flow Similarity Loss:** This loss ensures the motion of the layer $i$ is uncorrelated with the rest of the video layers. We define the loss as cosine similarity between the VGG embeddings of the masked flow-RGB of layer $i$ with the rest of the layers. The illustration for this loss is depicted in Fig. 4. We can mathematically write this loss as follows,

$$\mathcal{L}_{\text{Fsim}} = \frac{\phi(M_t^i \odot F^{\text{RGB}}) \cdot \phi((1 - M_t^i) \odot F^{\text{RGB}})}{\|\phi(M_t^i \odot F^{\text{RGB}})\| \|\phi((1 - M_t^i) \odot F^{\text{RGB}})\|}.$$ (9)

Here, $\phi(\cdot)$ in the Eqn. 9 denotes the embeddings obtained by the pre-trained VGG network trained on ImageNet. $F^{\text{RGB}}$ is the RGB image of flow estimate from frame $t-1$ to $t$.

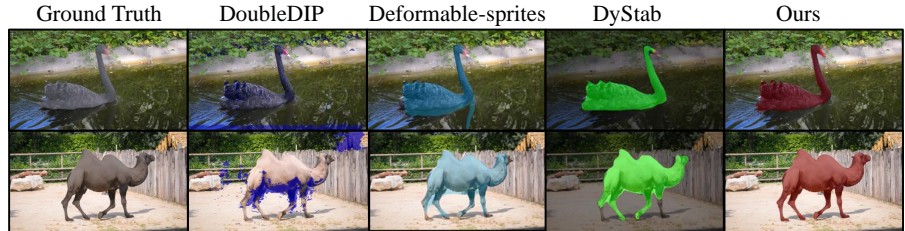

Figure 5: **Qualitative evaluation on VOS benchmark:** We compare the mask from our method with the baselines on DAVIS-16 (32) dataset. We compare our method against the baselines DS (58) and DyStab (56). Please note the above masks (exception: DoubleDIP) for baselines are pre-computed masks provided by the authors of the papers.

Table 2: **Quantitative evaluation on VOS benchmarks:** For the evaluation of our method, we calculate the IOU score ($\mathcal{J}$) between the generated masks and the ground truth mask. Also, blue **(bold)** denotes the best score. Please note that the DyStab is trained using a video dataset, while all the other methods require only a test time video sequence to segment the object from the video.

| Model | Flow model | DAVIS (32) | FBMS (47) |
|---|---|---|---|
| DoubleDIP (15) | - | 24.6 | - |
| ARP (21) | CPMFlow (18) | 76.2 | 59.8 |
| CIS (55) | PWCNet (46) | 71.5 | 63.5 |
| MG (54) | RAFT (48) | 68.3 | 53.1 |
| DS (58) | RAFT (48) | 79.1 | 71.8 |
| DyStab* (56) | PWCNet (46) | 80.0 | **73.2** |
| Ours | RAFT (48) | **81.1** | **73.2** |

**Mask Loss:** This loss ensures binarization in the generated layer mask. We utilized the mask loss put forth by Gandelsman et al. (2019). We can mathematically write this loss as follows,

$$\mathcal{L}_{\text{Mask}} = \sum_{L,T} \left( |M_t^i - 0.5| \right)^{-1}. \tag{10}$$

**Reconstruction layer Loss:** This loss ensures the appearance of the layer $i$ is a segment of the original input video. We define the loss as an L1 loss between the masked RGB image prediction of layer $i$ and the masked RGB image prediction of the original video. We can mathematically write this loss as follows,

$$\mathcal{L}_{\text{layer}} = \|M_t^i \odot X_t - M_t^i \odot \hat{X}_t\|. \tag{11}$$

Here, $X_t$ is the RGB frame in the video sequence at time-step $t$. $M_t^i$ is the alpha map for the $i^{\text{th}}$ object layer and $t^{\text{th}}$ frame.

**Final Modified Loss:** We combine all the losses together into a final loss term as follows,

$$\begin{aligned} \mathcal{L}_{\text{UVOS}} = & \lambda_{\text{rec}} \mathcal{L}_{\text{rec}} + \lambda_{\text{Fsim}} \mathcal{L}_{\text{Fsim}} + \lambda_{\text{layer}} \mathcal{L}_{\text{layer}} \\ & + \lambda_{\text{warp}} \mathcal{L}_{\text{warp}} + \lambda_{\text{Mask}} \mathcal{L}_{\text{Mask}}. \end{aligned} \tag{12}$$

We leverage our final modified loss objective, given by Eqn. 12, to perform unsupervised video object segmentation. For this task, we want to split an input video into foreground and background layers. Hence, we initialize 2 RGBnets ($f_{\text{RGB}}$) and one $\alpha$net ($f_\alpha$). Weights for all these modules are optimized using the test sequence only. The illustration of this is shown in the Fig. 4. Because we are dealing with a decomposition of the input video into two layers, i.e., as $L = 2$, we modify Eqn. 6 as follows,

$$X_t = M_t \odot f_{\text{RGB}}^1(X_t) + (1 - M_t) \odot f_{\text{RGB}}^2(X_t) \quad \forall t \in (1, n]. \tag{13}$$

All the loss functions in the Eqn. 12 are modified with the value of $L = 2$. We fit our model parameters using loss function given by Eqn. 12. We then use the learned $\alpha$-net to generate a segmentation mask for the input video. The predicted segmentation masks for each timestamp can be obtained by $M_t = f_\alpha(F_{t-1 \to t}^{\text{RGB}})$. where, $F_{t-1 \to t}^{\text{RGB}}$ is the RGB image of flow estimate from frame $t - 1$ to $t$.

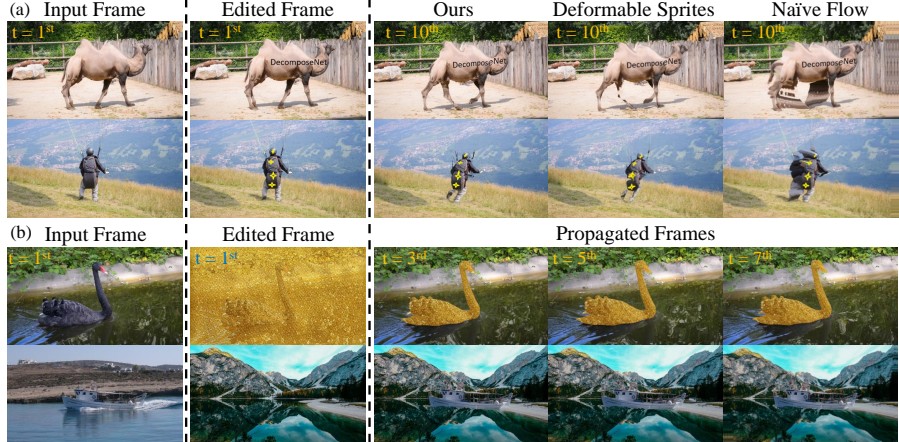

Figure 6: **Edits propagation:** First column in the figure represents the input frame to be edited. Second column represents the edits performed. Next columns in Fig. (a) represent the qualitative comparison of propagated edits with baselines, while Fig. (b) represents propagated edits throughout the video. From Row 1, 2 - it can be observed that the quality of our framework's propagated edited decals is much better as compared to the baselines. The legs of the camel(Row 1) and parachute bag shape(Row 2) are preserved in our method's edit propagation. Row 3 - It can be observed that consistent edits in a particular layer can be performed throughout the video. In this case, we show an example of style transfer being performed on the swan. Row 4 - we can selectively replace the background layer and propagate the edits through the video.

We evaluate our method for unsupervised video object detection on the DAVIS-16 dataset (Perazzi et al., 2016) and compare our model against the existing baselines. A qualitative comparison between our approach and the baselines can be seen in Fig. 5. It can be observed from Fig. 5 that our method performs better than the existing methods in terms of segmenting the most salient object from the scene. Table. 2 presents a quantitative evaluation of our approach. Our method outperforms the existing inference-time optimized baselines in the unsupervised video object segmentation setting. DoubleDIP is the worst-performing baseline in our evaluation.

We also perform an ablation study of our framework and refer our readers to the supplementary materials.

## 5.1 COHERENT EDITS PROPOGATION

With our approach, we are able to perform downstream applications like coherently editing the foreground and background object layers in a scene. It can be observed from Fig. 6 the type of edits we can perform on one layer in some keyframe, which can then be propagated throughout the video. The distinct advantage of our method over other such methods is that the user can perform the edit on any keyframe which has a better view of the object. This makes it easier for the user to make the edit instead of making it in one global atlas for the object layer with a distorted single view. In Fig. 6, we perform different edits such as add decals or modify a keyframe with style transfer Park et al. (2020) or swap the background. These edits are then consistently propagated using a combination of flow warping operation and composition using the obtained masks. We refer readers to our webpage in the supplementary material for video results.

## 6 VIDEO DEHAZING

Videos captured in outdoor settings are often degraded by a scattering medium (e.g., haze, fog, underwater scattering). The degradation is such that it grows with the depth of the scene. Typically, we can write a hazy video $\{X_t\}_1^T$ as put forth by papers (He et al., 2010; Gandelsman et al., 2019; Fattal, 2008; Narasimhan & Nayar, 2000):

$$X_t = \alpha \odot \text{Clr}(X_t) + (1 - \alpha) \odot A_t, \tag{14}$$

where $A_t$ is the Airlight map (A-map), $\text{Clr}(X_t)$ is the haze-free image, and $\alpha$ here is the transmission (t-map).The goal of video dehazing is to recover the clear underlying video $\{\text{Clr}(X_t)\}_1^T$ from a hazy input video $\{X_t\}_1^T$,i.e., recover the input video that would have been captured on a clear day with

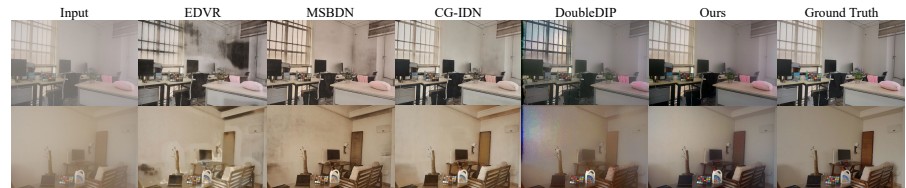

Figure 7: **Qualitative evaluation on Video Dehazing benchmark:** We compare the dehazed result of our method with the baselines on the REVIDE (61) dataset. We compare our method against both the image and video baselines. MSBDN (10) method is an image-based baseline while EDVR (53), DoubleDIP (15) and CG-IDN (61) are video baselines. Please note that both the EDVR and CG-IDN methods are trained using the training data of the REVIDE dataset. In contrast, our approach operates directly on the hazy video sequence. As can be seen from the figure, our approach to dehazing produces a color palette in the dehazed frame that closely resembles the color palette in the ground truth frame.

Table 3: **Quantitative evaluation on Video Dehazing benchmarks:** To evaluate our method, we calculate the Avg PSNR and SSIM scores between the generated haze-free video and ground truth video. For all baselines and our method, the score is calculated on the REVIDE dataset (61).

| Metrics | EDVR (53) | DoubleDIP (15) | MSBDN (10) | CG-IDN (61) | Ours |
|---|---|---|---|---|---|
| PSNR | 21.22 | 17.32 | 22.01 | 23.21 | **24.83** |
| SSIM | 0.8707 | 0.7942 | 0.8759 | 0.8836 | **0.9164** |

good visibility conditions. We treat the dehazing problem as a video decomposition problem, where one layer is the haze-free video and the other layer is the Airlight map. This Airlight map is obtained using the standard methodology proposed in Bahat & Irani (2016) and kept as a fixed (non-trainable) layer. Further, we model the transmission t-maps with our $\alpha$-net model. In summary, to find the dehazed video from the given input video, we initialize one RGBnet, one $\alpha$-net, and use the Eqn. 14 for reconstruction loss.

To optimize the model parameters, we optimize the objective function given by Eqn. 1. We evaluate our method for the video dehazing task on the REVIDE dataset (Zhang et al., 2021b) and compare our model against the existing state-of-the-art baselines. It is important to note that the REVIDE dataset consists of real-world captured hazy RGB video and clean video pairs. REVIDE dataset has a total of 48 sequence pairs (hazy seq, gt seq). We choose both image-based and video-based approaches as baselines. MSBDN (Dong et al., 2020) is an image based dehazing baseline, while DoubleDIP (Gandelsman et al., 2019), EDVR (Wang et al., 2019) and CG-IDN (Zhang et al., 2021b) are the video based baseline. EDVR and CG-IDN approaches are trained using the videos of the REVIDE dataset; hence, these methods act as perfect baselines.

A qualitative evaluation can be seen in Fig. 7. Note that only ours and DoubleDIP approaches do not require any task-specific training data. Both approaches are optimized directly on the inference-time video sequence instead of other approaches that have to be trained using external data. Table. 3 represents the quantitative evaluation of our approach. It can be observed from the table that our method outperforms the existing baselines by a significant margin in the video dehazing task. DoubleDIP performs worst among the baselines in our quantitative evaluation. We also refer the readers to an important ablation study detailed in Sec. E where we examine the importance of FlowRGB utilization for relighting and dehazing tasks.

# 7 CONCLUSION

We have demonstrated in the paper how the layered decomposition could be applied to a wide variety of downstream tasks. It is important to note that our model does not require any explicit training, i.e., we optimize the parameters using the test sequence itself. Additionally, in this paper, we achieve state-of-the-art performances in downstream tasks such as UVOS (among the other test time optimization techniques), dehazing, and relighting. We also demonstrated that the targeted edits propagation can also be performed as a downstream task of UVOS. With the help of semantic or perception cues, we believe our framework can also be extended for monocular depth estimation, semantic segmentation, and other high-level vision tasks.

**Acknowledgements.** This project was partially funded by NSF CAREER Award (2238769) to AS. We also thank Harsh Shrivastava, Saurabh Singh, Matthew Walmer, Soumik Mukhopadhyay, Shirley Huang and Bo He for providing feedback on the manuscript.

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

Table 4: **RGBnet:** Each convolution is followed by batch normalization and a leaky rectified linear unit (LeakyReLU; negative slope = 0.2), except for the last layer of the encoder and decoder. In the last layer of the encoder and decoder, we use the sigmoid activation function. We utilize Pytorch for our implementation.

| Module | Type | Kernel Size | Num Inputs | Num Outputs | Stride | Padding |
|---|---|---|---|---|---|---|
| | Conv2D | $3 \times 3$ | 3 | 64 | 1 | 1 |
| | Conv2D | $3 \times 3$ | 64 | 64 | 1 | 1 |
| | Maxpool2D | $2 \times 2$ | 64 | 64 | 2 | 0 |
| | Conv2D | $3 \times 3$ | 64 | 96 | 1 | 1 |
| | Conv2D | $3 \times 3$ | 96 | 128 | 1 | 1 |
| Encoder | Maxpool2D | $2 \times 2$ | 128 | 128 | 2 | 0 |
| | Conv2D | $3 \times 3$ | 128 | 128 | 1 | 1 |
| | Conv2D | $3 \times 3$ | 128 | 128 | 1 | 1 |
| | Conv2D | $3 \times 3$ | 128 | 128 | 1 | 1 |
| | Maxpool2D | $2 \times 2$ | 128 | 128 | 2 | 0 |
| | Conv2D | $4 \times 4$ | 128 | 96 | 1 | 0 |
| | ConvTranspose2d | $4 \times 4$ | 96 | 128 | 1 | 0 |
| | Bicubic upsample | (scale =2) | - | - | - | - |
| | Conv2D | $3 \times 3$ | 128 | 128 | 1 | 1 |
| | Conv2D | $3 \times 3$ | 128 | 128 | 1 | 1 |
| | Conv2D | $3 \times 3$ | 128 | 128 | 1 | 1 |
| Decoder | Bicubic upsample | (scale =2) | - | - | - | - |
| | Conv2D | $3 \times 3$ | 128 | 96 | 1 | 1 |
| | Conv2D | $3 \times 3$ | 96 | 64 | 1 | 1 |
| | Bicubic upsample | (scale =2) | - | - | - | - |
| | Conv2D | $3 \times 3$ | 64 | 64 | 1 | 1 |
| | ConvTranspose2D | $3 \times 3$ | 64 | 3 | 1 | 1 |

## A    NETWORK ARCHITECTURE FOR RGBNET AND $\alpha$-NET

RGBnet is a fully convolutional network. It is designed in such a way as to handle frames of all resolutions. The input to this network is RGB frame at time $t$. We utilize a U-net architecture for this network without the skip connections. More details about the architecture can be found in Table. 4. Similarly, more details about the $\alpha$-net can be found in Table. 5.

## B    OPTIMIZATION DETAILS

To optimize our model, we use a single Nvidia A6000 GPU with 48G memory to process a single video at a time of resolution 856x480. We optimize the module ($f_{\mathrm{RGB}}(\cdot)$ and $f_\alpha(\cdot)$) weights using the entire test sequence with the *Adam optimizer* at a learning rate in the range of [0.00002, 0.002]. Because our approach is non-recurrent, it can be scaled up by parallelizing over multiple GPUs, which speeds up the process for longer sequences. We achieve a processing rate of 60 frames/min for dehazing and relighting and 40 frames/min for VOS with one salient object. However, we can speed up the VOS task by reducing the dimension from 856x480 to 224x128 and then running the VOS task pipeline. This results in a 16x speed-up of the pipeline, increasing the processing rate to 640 frames/min without affecting the efficacy of our method. We use 100 epochs for a 60-frame sequence in VOS and 60 epochs for dehazing and relighting. Additionally, we conducted our experiments using multiple random initializations of $f_{\mathrm{RGB}}(\cdot)$ and $f_\alpha(\cdot)$ modules and observed consistent quantitative results across all runs.

### B.1    HYPER PARAMETER SETTING

**Unsupervised Video Object Segmentation:** For getting a good performance on the UVOS task we utilize the following weights for the different losses; $\lambda_{\mathrm{rec}} = 1$, $\lambda_{\mathrm{Fsim}} = 0.001$, $\lambda_{\mathrm{layer}} = 1$, $\lambda_{\mathrm{warp}} = 0.01$ and $\lambda_{\mathrm{Mask}} = 0.01$.

Table 5: $\alpha$-**net:** Each convolution layer is followed by batch normalization and a leaky rectified linear unit (LeakyReLU; negative slope = 0.2), except for the last layer of the encoder and decoder. In the last layer of the encoder and decoder, we use the sigmoid activation function. We utilize Pytorch for our implementation.

| Module | Type | Kernel Size | Num Inputs | Num Outputs | Stride | Padding |
|---|---|---|---|---|---|---|
| Encoder | Conv2D | $3 \times 3$ | 3 | 64 | 1 | 1 |
| | Conv2D | $3 \times 3$ | 64 | 64 | 1 | 1 |
| | Maxpool2D | $2 \times 2$ | 64 | 64 | 2 | 0 |
| | Conv2D | $3 \times 3$ | 64 | 96 | 1 | 1 |
| | Conv2D | $3 \times 3$ | 96 | 128 | 1 | 1 |
| | Maxpool2D | $2 \times 2$ | 128 | 128 | 2 | 0 |
| | Conv2D | $3 \times 3$ | 128 | 128 | 1 | 1 |
| | Conv2D | $3 \times 3$ | 128 | 128 | 1 | 1 |
| | Conv2D | $3 \times 3$ | 128 | 128 | 1 | 1 |
| | Maxpool2D | $2 \times 2$ | 128 | 128 | 2 | 0 |
| | Conv2D | $4 \times 4$ | 128 | 96 | 1 | 0 |
| Decoder | ConvTranspose2d | $4 \times 4$ | 96 | 128 | 1 | 0 |
| | Bicubic upsample | (scale =2) | - | - | - | - |
| | Conv2D | $3 \times 3$ | 128 | 128 | 1 | 1 |
| | Conv2D | $3 \times 3$ | 128 | 128 | 1 | 1 |
| | Conv2D | $3 \times 3$ | 128 | 128 | 1 | 1 |
| | Bicubic upsample | (scale =2) | - | - | - | - |
| | Conv2D | $3 \times 3$ | 128 | 96 | 1 | 1 |
| | Conv2D | $3 \times 3$ | 96 | 64 | 1 | 1 |
| | Bicubic upsample | (scale =2) | - | - | - | - |
| | Conv2D | $3 \times 3$ | 64 | 64 | 1 | 1 |
| | ConvTranspose2D | $3 \times 3$ | 64 | 1 | 1 | 1 |

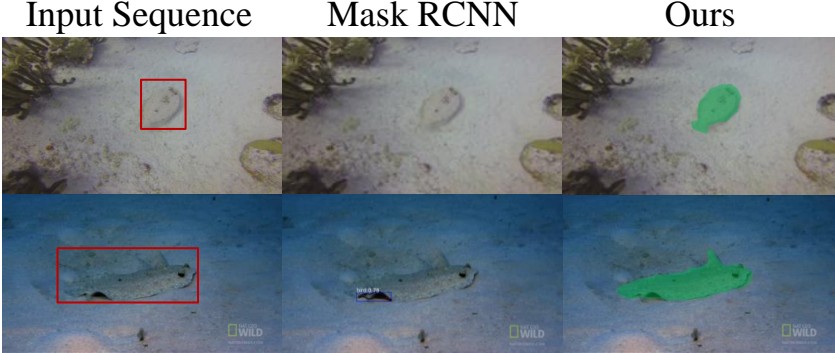

Figure 8: **Comparison with pretrained image segmentation model:** The examples presented in the figure above are samples from the MoCA dataset (Lamdouar et al., 2020). In comparison to MaskRCNN (He et al., 2017), a pretrained image segmentation model trained on the COCO dataset, our model is more robust in handling unfamiliar objects in the scene. The figure shows some examples where MaskRCNN fails to segment the unfamiliar object correctly, while our approach produces much better segmentation results.

**Unsupervised Video Dehazing:** For getting a good performance on the Dehazing task we utilize the following weights for the different losses; $\lambda_{\text{rec}} = 1$ and $\lambda_{\text{warp}} = 0.02$.

**Unsupervised Video Relighting:** For getting a good performance on the Relighting task we utilize the loss weights similar to the Dehazing task, i.e., $\lambda_{\text{rec}} = 1$ and $\lambda_{\text{warp}} = 0.02$.

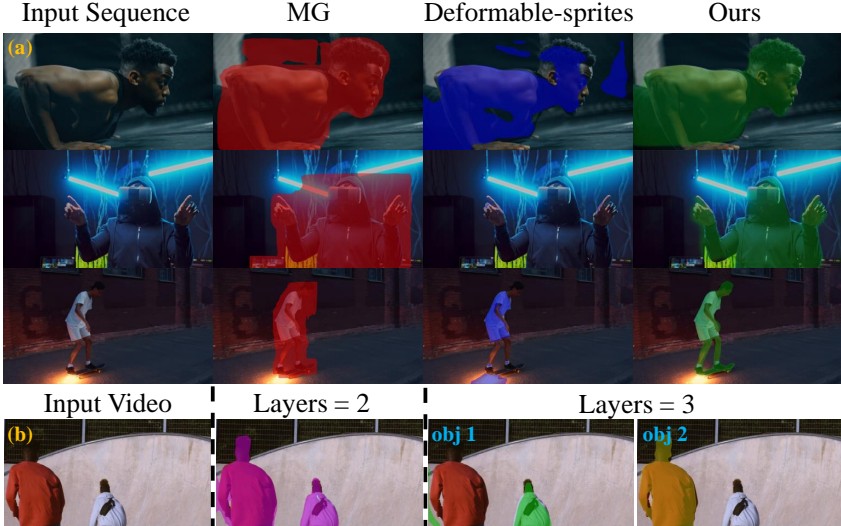

Figure 9: **(a) Segmentation in variable lighting conditions throughout the scene.** We compare the mask from our method with the baselines Deformable sprites (Ye et al., 2022) and MG (Yang et al., 2021a) on frames of three video sequences. It can be observed from the above figure methods such as Deformable-sprites and MG are more prone to lighting changes as opposed to our method. **(b) Decomposition into multiple layers** We demonstrate an example where the number of layers $L$ is set to both $L = 2$ and $L = 3$. $L = 2$ results in foreground and background layers separation, while $L = 3$ results in the separation of individual person in different object layers.

## C  UVOS

### C.1  UVOS - SCENE WITH IRREGULAR OBJECTS

In Fig. 8, we demonstrate that our method doesn't mandate the object to belong to a common category to achieve the correct decomposition. This is in stark contrast to other methods like that of (Kasten et al., 2021), which heavily relies on MaskRCNN or a user-input mask for video decomposition. This flexibility provides our approach with an advantage, allowing for precise video foreground (FG) and background (BG) edits and ensuring seamless propagation throughout the sequence.

### C.2  UVOS - LIGHTING CHANGES

It's notable that the efficacy of deformable sprites (Ye et al., 2022) tends to deteriorate in video sequences where the appearance of an object undergoes alterations due to shifts in lighting. This decrement in performance emerges primarily because deformable sprites attempt to encapsulate an object's universal appearance using a singular canonical RGB texture. In contrast, our framework operates by modeling video decomposition at an individual frame level. Such an approach amplifies the resilience of the object segmentation process against lighting modifications, a fact that is evident from Fig. 9.(a).

### C.3  UVOS - DECOMPOSITION INTO MULTIPLE LAYERS

Our model's capabilities are not confined to a binary layer decomposition (Foreground layer and Background layer). As depicted in Fig. 9.(b), upon configuring the layer parameter to 2, our model precisely segregates the foreground from the background. Intriguingly, when the layer parameter is adjusted to 3, the model successfully isolates each person into their distinct layer while maintaining a separate layer for the background.

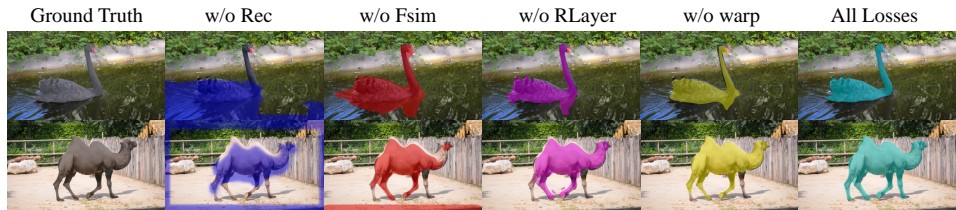

Figure 10: **Ablation Study:** We study the effect of using only a subset of the loss function for the video object segmentation task.

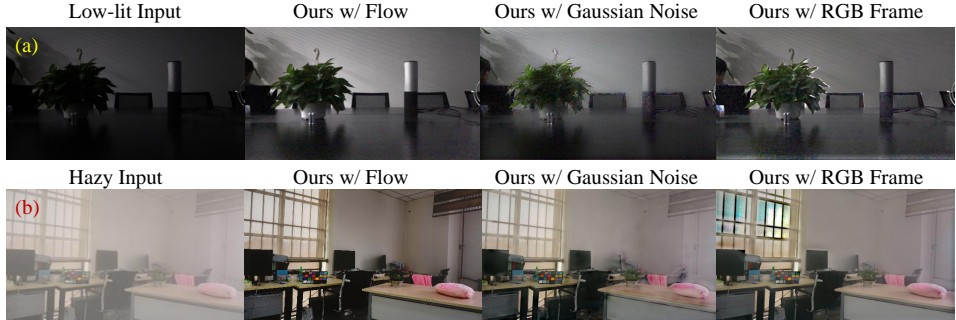

Figure 11: **Ablation Study:**This figure presents the results of an ablation study where we analyzed the impact of using FlowRGB, Gaussian Noise, and RGB frame as input to the $\alpha$-net. Row (a) shows a qualitative comparison for relighting tasks, where we evaluated three different settings for the $\alpha$-net. Row (b) presents a comparison for the dehazing task.

## C.4 UVOS - ABLATION STUDY

We performed ablation experiments to study the effect of losses for our unsupervised video object segmentation method. We report the quantitative results on the DAVIS-16 dataset in Table. 6. In Fig. 10, we show a qualitative comparison between architectures optimized with different subsets of losses given by Eqn. 12. It can be observed from Fig. 10 that removing the reconstruction loss drastically reduces the ability of our method to segment the object of interest in the scene. This is also reflected in the quantitative evaluation given by Table. 6. Dropping the $\mathcal{L}_{\text{Fsim}}$ reduces our model's ability to separate the motion of the foreground object from the background scene. Optimizing Eqn. 12 without the $\mathcal{L}_{\text{layer}}$ results in a performance drop as the obtained object mask lacks the appearance grouping provided by the addition of the layer loss. Lastly, dropping the $\mathcal{L}_{\text{warp}}$ reduces the temporal consistency of the masks throughout the sequence.

## C.5 COMPLETE VOS EVALUATION FOR DAVIS-16

We include the supervised VOS approaches along with the unsupervised approaches for the completeness of the evaluation. It is important to note that unsupervised in the context of this paper implies that no human annotation is required either for the training or testing process of the approach. Table. 8 reflects the evaluation of the baselines on the DAVIS-16 dataset for VOS task.

## D INTUITION FOR DEHAZING/RELIGHTING TASKS

**Dehazing:** Previous studies, such as Bahat & Irani (2016), have shown that small image patches are recurrent in natural images, but this recurrence is disrupted in non-ideal conditions such as haze/noise. Gandelsman et al. (2019) established this patch recurrence property could be exploited using a coupled DIP (Ulyanov et al., 2018). When a combination of multiple DIPs is used to reconstruct a frame, those DIPs tend to "split" a frame such that the patch distribution of each DIP output follows high patch recurrence. Hence, when we use a coupled version of U-nets in our framework to

Table 6: **Ablation Study:** We utilize the DAVIS-16 dataset for performing the ablation study. We calculate the IOU score ($\mathcal{J}$) between the generated mask and the ground truth mask.

|  | w/o $\mathcal{L}_{\text{rec}}$ | w/o $\mathcal{L}_{\text{Fsim}}$ | w/o $\mathcal{L}_{\text{layer}}$ | w/o $\mathcal{L}_{\text{warp}}$ | All Losses |
|---|---|---|---|---|---|
| Davis | 52.6 | 67.3 | 72.8 | 78.5 | **81.1** |

Table 7: **Quantitative evaluation on Dehazing/Relighting ablations:** This Table presents the results of quantitative evaluations conducted on dehazing and relighting tasks using three different settings of our method. To assess the performance of our approach, we measured the average PSNR and SSIM scores between the processed videos and ground truth videos. For dehazing evaluations, we used the REVIDE (Zhang et al., 2021b) dataset, and for relighting evaluations, we used the SDSD (Wang et al., 2021) dataset.

| Metrics | Task | **Ours w/ FlowRGB** | Ours w/ Gaussian Noise | Ours w/ RGB |
|---|---|---|---|---|
| PSNR | Dehazing | **24.83** | 16.79 | 16.84 |
| SSIM | Dehazing | **0.9164** | 0.7768 | 0.7721 |
| PSNR | Relighting | **27.92** | 23.74 | 24.52 |
| SSIM | Relighting | **0.9046** | 0.7221 | 0.7342 |

reconstruct the haze frame, it always converges into a natural image with repetitive patches (termed as a haze-free image) and a non-linear airlight map.

**For the relighting task**, we begin by defining a manifold for the relit frame, which is based on the gamma correction formulation as delineated in Eqn. 5. Our architectural choice incorporates convolutional U-Nets, leveraging their inherent property to enhance self-similarity in the produced outputs. Notably, natural images, particularly those captured in optimal conditions, consistently exhibit the patch recurrence property. When we employ the U-Net for optimization, its output (termed as 'relit' frame) reliably gravitates towards a natural image marked by repetitive patches. Concurrently, the system fine-tunes and selects the appropriate parameters for $\gamma^{-1}$ and Tmaps ($1/A_t$). A salient feature of the U-Net's output is its pronounced patch recurrence, which, in turn, translates to a substantial reduction in noise levels for the relit frame. This inherent noise reduction alleviates the need for a subsequent, explicit denoising process.

# E    ABLATION FOR DEHAZING AND RELIGHTING TASKS

We perform ablation studies of our framework for dehazing and relighting tasks. We evaluate our method utilizing three different settings, namely

- First setting, where we provide flow-RGBs as an input to $\alpha$-net (ours VDP prior setting).

- Second setting, here, we provide a Gaussian Noise as an input to $\alpha$-net.

- Third setting, in which we provide an RGB frame as an input to $\alpha$-net.

We report the quantitative evaluation on REVIDE dataset for dehazing and SDSD dataset for relighting tasks. Please refer to Table. 7 and Fig. 11 for the findings. It can be observed that using flow-RGBs as input to $\alpha$-net provides significant gain over the other settings. We attribute this behavior to information obtained from neighboring frames that help the model to refine the Tmaps further and produce results with higher fidelity.

It is important to note that in our experiments, we initially attempted to employ raw (x,y) offsets as feature maps for input to the $\alpha$-Net. However, this approach exhibited suboptimal normalization performance. Conversely, the colorization of optical flow effectively normalizes flow-rgb values within the range [0,1], proving to be a more suitable input for the $\alpha$-Net.

Table 8: **Quantitative evaluation on VOS benchmarks:** This Table presents the results of a quantitative evaluation of our method on VOS benchmarks. To assess the performance of our approach, we calculated the IOU score ($\mathcal{J}$) between the generated masks and the ground truth mask on the DAVIS dataset (Perazzi et al., 2016). The scores were computed for all baselines and our method. It is important to note that the methods listed above the line require no annotated data for performing the VOS task, while the methods mentioned below the line require annotated data for training. The best score in each category is highlighted in blue (**bold**).

| Model | Annotations | No. of Annotations | DAVIS (Perazzi et al., 2016) | FBMS (T.Brox & J.Malik, 2010) |
|---|---|---|---|---|
| DoubleDIP (Gandelsman et al., 2019) | - | - | 24.6 | - |
| ARP (Koh & Kim, 2017) | - | - | 76.2 | 59.8 |
| CIS (Yang et al., 2019a) | - | - | 71.5 | 63.5 |
| MG (Yang et al., 2021a) | - | - | 68.3 | 53.1 |
| DS (Ye et al., 2022) | - | - | 79.1 | 71.8 |
| DyStab* (Yang et al., 2021b) | - | - | 80.0 | **73.2** |
| **Ours** | - | - | **81.1** | **73.2** |
| COSNet (Lu et al., 2020b) | ✓ | 17,000 | 80.5 | 75.6 |
| AnDiff (Yang et al., 2019b) | ✓ | 2,000 | 81.7 | - |
| EPONet (Faisal et al., 2019) | ✓ | 2,000+ | 80.6 | - |
| PDB (Song et al., 2018) | ✓ | 17,000 | 77.2 | 74.0 |
| LVO (Tokmakov et al., 2017) | ✓ | 2,000+ | 75.9 | 65.1 |
| MATNet (Zhou et al., 2020) | ✓ | 14,000 | 82.4 | **76.1** |
| DyStab (Yang et al., 2021b) | ✓ | 2,000 | **82.8** | 73.2 |

Table 9: **Quantitative evaluation of Video Edits Propagation:** For the evaluation of our method, we calculate the Avg PSNR and SSIM scores between the reconstructed videos and ground truth video sampled from the DAVIS-16 dataset 32. We use the code for the baselines that are publicly available to calculate the PSNR and SSIM scores. For all baselines and our method, the score is calculated by averaging the PSNR and SSIM value over a sequence length of 15 frames (frames: 2-16).

| Metrics | Naive Flow | Deformable Sprites (Ye et al., 2022) | Ours |
|---|---|---|---|
| PSNR | 22.54 | 27.50 | **29.73** |
| SSIM | 0.4527 | 0.9543 | **0.9726** |

## F    EDIT PROPAGATION

In the case of Salient Object Stylization/Background swapping, we utilize $X_0$ as the edited frame. The forward optical flow $F_{t \to t+1}$ is obtained using the original input video. Given a frame $X_t$ at time $t$ and the forward flow $F_{t \to t+1}$ from frame $t$ to frame $t + 1$, we can define the warped frame $X'_{t+1}$ using the following relation:

$$X'_{t+1}(u, v) = X_t(u + \Delta u, v + \Delta v) \tag{15}$$

Where $\Delta u$ and $\Delta v$ are the horizontal and vertical displacements, respectively, from the forward flow $F_{t \to t+1}$ at pixel location $(u, v)$.

Furthermore, let $M_{t+1}$ be a binary mask at time $t + 1$ where its values are either 0 or 1. The prediction $\hat{X}_{t+1}$ for a pixel at location $(u, v)$ is then obtained by:

$$\hat{X}_{t+1}(u, v) = M_{t+1}(u, v) \odot X_{t+1}(u, v) + (1 - M_{t+1}(u, v)) \odot X'_{t+1}(u, v) \tag{16}$$

This formulation suggests that the prediction $\hat{X}_{t+1}$ is a weighted combination of the actual frame $X_{t+1}$ and the warped frame $X'_{t+1}$, where the weighting is determined by the mask $M_{t+1}$.

### F.1    EDITS - EVALUATION

To quantitatively evaluate the effectiveness of our approach for propagating coherent edits, we conducted a proxy experiment. Rather than editing the frame, we applied the warping operations without

modifying the original frame. We then evaluated the quality of the reconstructed warped frames and compared our results against the baselines. The findings of this experiment are presented in Table. 9, which shows that our method outperformed all of the other baselines.

## G  LIMITATIONS OF OUR APPROACH

Our method has four limitations that must be considered. First, the accuracy of the decomposition is dependent on the effectiveness of the optical flow estimator network (RAFT). This means that any inaccuracies or limitations of the network will directly impact the quality of the results. Second, the number of separable layers is a hyperparameter that must be determined prior to implementation. Third, If the number of separable layers is set to N, then 2N independent $f_{\text{RGB}}$ and $f_\alpha$ need to be initialized, which can significantly increase computational costs. Finally, it's worth noting that our method is a test-time optimization technique and may be slower than its trained counterparts. However, as mentioned earlier, our model can be sped up by resizing the frame size for the VOS task or by utilizing multiple GPUs for optimization. These limitations must be taken into consideration when implementing our method in practice.

## H  OPTICAL FLOW

Optical flow pertains to the pattern of visible motion of objects, surfaces, and edges within a visual scene. It provides an estimation of the motion of objects between two consecutive frames, which results from the relative movement between the object and the camera. **Unless mentioned otherwise we use pretrained RAFT (Teed & Deng, 2020) model to predict the optical-flow between the two consecutive frames.**

Given two successive frames $X_t(u, v)$ and $X_{t+1}(u, v)$, the goal of optical flow is to identify a displacement vector $(\Delta u, \Delta v)$ for each pixel $(u, v)$ such that:

$$X_t(u, v) = X_{t+1}(u + \Delta u, v + \Delta v) \tag{17}$$

Here, $\Delta u$ and $\Delta v$ represent the horizontal and vertical displacements, respectively.

## I  OPTICAL FLOW WARPING OPERATION

Optical flow warping denotes the application of the computed optical flow vectors to "warp" or "remap" one image in order to predict the subsequent image in the sequence. For each pixel in the present frame, its new position in the next frame is forecasted via the optical flow, transferring the intensity to this new location.

Given an frame $X_t$ and its optical flow $F_{t \to t+1} = (\Delta u, \Delta v)$, the warped image $X'_t$ for the next time frame $t + 1$ can be expressed as:

$$X'_t(u + \Delta u, v + \Delta v) = I_t(u, v) \tag{18}$$

## J  OPTICAL FLOW RGB REPRESENTATION

To visualize optical flow vectors, one can represent them in an RGB format. The 2D optical flow vectors $(\Delta u, \Delta v)$ are first converted into polar coordinates $(r, \theta)$. The hue in the HSV color space corresponds to the motion direction, while the intensity or value indicates the motion magnitude.

To transition from $(\Delta u, \Delta v)$ to $(r, \theta)$:

$$\theta = \arctan 2(\Delta v, \Delta u) \tag{19}$$
$$r = \sqrt{\Delta u^2 + \Delta v^2} \tag{20}$$

Subsequently, $\theta$ and $r$ are proportionally scaled and mapped to their respective HSV values. Afterward, this HSV image can be converted into RGB for effective visualization. Note that to enhance visualization, there's often an imposed limit on the maximum value of $r$. Thus, any motion exceeding this threshold is typically visualized at maximum brightness.

