# OpenReview forum: "Video Decomposition Prior: Editing Videos Layer by Layer"
_ICLR.cc/2024/Conference — ICLR 2024 poster_

### Official Review · Reviewer_6Tks · 2023-10-21

**Soundness:** 3 good
**Presentation:** 3 good
**Contribution:** 2 fair
**Rating:** 6
**Confidence:** 5

**Summary:**

The paper proposes a new framework that decomposes a video into multiple multiple RGB layers and associated opacity levels for various video editing tasks.
The proposed framework uses two neural network modules, RGB-net and α-net, to predict RGB layers and opacity/transmission layers for each video frame.
These layers are then composited to reconstruct the input video or achieve the desired effects during optimization.
The paper conducted experiments on video relighting, dehazing, and VOS tasks and achieves superior results compared to some existing baselines.

**Strengths:**

- Writing is overall clear.
- Various tasks are conducted and better performance w.r.t to some baseline is reported.
- No pre-training is required (test-time optimization).

**Weaknesses:**

- This paper proposes to unify various downstream tasks by video decomposition. However, the formulation of the video relighting task in this paper uses one RGB layer to represent the enhanced video and an alpha-layer to represent gamma correction. This formulation does not have any meaning of “decomposition”. On the other hand, this paper carefully designs different decomposition definitions and corresponding constraints for different downstream tasks, but these constraints and decomposition definitions seem to lack a unified and general formulation. Therefore, I felt such kind of “unification” seems too artificial and forced.

- The proposed formulation is not new (or at least has very few novelty).
Overall, the proposed formulation can be summarized as a "data-term + prior" approach, which is a rather common formulation in optimization-based approach for image / video synthesis tasks.
Using neural networks as an implicit prior for data term is not new (as also mentioned in the paper - "These approaches have highlighted the importance of formulating a loss function,
combined with the optimization of neural network parameters") and the proposed paper extends this idea into other tasks.
Yet, the exact formulation designed for each task is dedicated but common (e.g., gamma-correction curves [1] for relighting; alpha-blending [2,3] for video-segmentation, dark-channel prior for dehazing).

- Missing baselines for UVOS tasks: The paper compares with both pre-trained methods and test-optimization methods. For pre-trained methods, there are massive VOS methods [4,5,6] that has far better performance than compared methods in this paper; For test-optimization methods, there are also other video decomposition methods that targets this task [2,3].

[1] Zhang, Mohan, et al. "RT-VENet: a convolutional network for real-time video enhancement." Proceedings of the 28th ACM International Conference on Multimedia. 2020.

[2] Kasten, Yoni, et al. "Layered neural atlases for consistent video editing." ACM Transactions on Graphics (TOG) 40.6 (2021): 1-12.

[3] Gu, Zeqi, et al. "Factormatte: Redefining video matting for re-composition tasks." ACM Transactions on Graphics (TOG) 42.4 (2023): 1-14.

[4] Cheng, Ho Kei, Yu-Wing Tai, and Chi-Keung Tang. "Rethinking space-time networks with improved memory coverage for efficient video object segmentation." Advances in Neural Information Processing Systems 34 (2021): 11781-11794.

[5] Yan, Kun, et al. "Two-shot Video Object Segmentation." Proceedings of the IEEE/CVF Conference on Computer Vision and Pattern Recognition. 2023.

[6] Cheng, Ho Kei, and Alexander G. Schwing. "Xmem: Long-term video object segmentation with an atkinson-shiffrin memory model." European Conference on Computer Vision. Cham: Springer Nature Switzerland, 2022.

**Questions:**

- While the proposed method does not require pre-training, it needs to optimization on each input sequence which takes time. How long does it take for optimizing a video sequence?

- Is there a common insight or general guidance on applying this framework to downstream tasks? For example, what if applying this framework to tasks like denoising or super resolution?

---

> ### Author Response · Authors · 2023-11-11
>
> We thank the reviewer for taking their time out of a busy schedule and evaluating our submission. We are glad that reviewer 6Tks found our paper well-written. We greatly appreciate that reviewer 6Tks finds the test-time optimization factor to be a strength of our approach. We are happy to note that the reviewer found our approach performance better across different tasks. Before we address the specific weaknesses and questions of the reviewer, **we highly recommend R-6Tks to please go through the webpage of video results**, which was part of the supplementary.
>
>
> **Relighting Decomposition:** We understand the reviewer’s concerns regarding the formulation of the relighting task. We would like to emphasize that the decomposition of a frame may not involve only additive decomposition($\alpha$-blending). What we explored in this work (one of our major novel contributions) is to perform multiplicative decomposition on frames of the video sequence. We derived one such multiplicative decomposition utilizing the gamma correction formulation for the frame relighting task. The interesting outcome of this decomposition is we are able to achieve SOTA performance on the relighting task without requiring any task-specific data. Not only that, we are also able to achieve high temporal coherence in the processed relit videos. Such qualitative comparison can be found in videos attached with the webpage(supp.).
>
> **Unified Framework:** We hope to change the reviewer’s viewpoint about the VDP framework being artificial. We have taken great pains to figure out this method.
>
> The core of our framework lies in its ability to adapt to different tasks while maintaining a consistent structure. Basically, each task consists of three integral components: prior knowledge (decomposition formulation), motion modeling (via $\alpha$-Net), and appearance modeling (via RGBnet). The **consistent architecture** of $\alpha$-Net and RGBnet establishes a **unified** way of extracting the motion and appearance features from a video, across different tasks. These intermediate motion and appearance features are then utilized with task-specific adaptations that are reflected in the decomposition formulations. Hence, the unifying factor across different tasks is the consistency in the architecture used to extract motion and appearance cues from the videos across all tasks.
>
>
> **Novelty Issues:** We claim novelty on two terms.  First, The relighting task formulation derived from gamma correction is novel for low-lit video enhancements and is unique to our work. Such formulation is not utilized in the work[1] mentioned as a reference by the reviewer.
>
> Second, The VDP framework’s way of modeling motion and appearance in a video is unique to our work(utilizing $\alpha$-Net and RGBnet). Specifically, to our knowledge, no prior test time optimization technique has utilized FlowRGBs to model the motion of the video, and it is unique to our methodology.
>
> With regards to the mentioned test-time methods[2,3], they are not at all designed for the VOS task but more for edit propagation tasks. These methods require per-frame masks obtained by image segmentation baselines. So, we do not see any overlap of these techniques with our framework apart from edit propagation tasks.
>
> We also would like to emphasize that maintaining spatio-temporal coherence is a big issue when it comes to tasks like dehazing and relighting, as can be seen from the qualitative results provided in the webpage(supplementary material). Our novelty also stems from the fact that our method can provide high-fidelity results with far superior spatio-temporal coherency. On top of this, we are able to achieve SOTA results for video dehazing and relighting tasks and achieve SOTA among the test-time optimization techniques in UVOS tasks. We hope this clarification will alleviate the reviewer’s concern about the novelty issues.
>
> **Missing VOS baselines:** In Table 2, we only compare with other test-time optimization baselines for a fair comparison as the setting of each methodology is exactly the same, i.e., no prior segmentation annotation is given to the model during training. We provide a comprehensive evaluation of the task UVOS task in Table 5 in the appendix because of the limited availability of space in the main paper. We will definitely add the remaining baselines[4,5,6] in there for a more comprehensive comparison. Additionally, if ICLR allows one extra page after acceptance, we would move those comparisons to the main paper.
>
> **Time taken:** We refer the reviewer to our appendix section C.
>
> **General Guidance:** We have explored three major downstream tasks like the VOS, dehazing, and relighting for videos. [1]* utilizes a test-time optimization technique for tasks like denoising, super-resolution, etc. Our framework can be naively extended to such tasks by utilizing similar loss functions; however, we leave this extension for future work.
>
> [1]* `Video dynamics prior’ - Neurips 2023

---

> > ### Comment · Reviewer_6Tks · 2023-11-18
> > **Further comments (1/2)**
> >
> > I appreciate the authors detailed response. Here are my further comments:
> >
> > > What we explored in this work (one of our major novel contributions) is to perform multiplicative decomposition on frames of the video sequence. We derived one such multiplicative decomposition utilizing the gamma correction formulation for the frame relighting task.
> >
> > I agree that multiplicative decomposition is suitable for relighting task. That being said, the exact form of decomposition (i.e., whether it is multiplicative, or additive, or any other form) is highly dependent on the exact task, which actually means that the proposed framework is highly task based but not sufficiently general. That dedicated design for relighting is technically reasonable; yet I found it hard for me to judge whether it is a "major novel" contribution.
> >
> > > Not only that, we are also able to achieve high temporal coherence in the processed relit videos. Such qualitative comparison can be found in videos attached with the webpage(supp.).
> >
> > The relighting quality looks great; the temporal coherence (as I played them again and again) looks very similar for both the proposed method, ZeroDCE++, and StableLLVE. Hence it is hard to say which one has "bettter" temporal coherency.
> > (BTW one interesting point I noted is that ZeroDCE++ produces also temporal stable results but with noisy outputs. From my understanding, the noisy-eliminated result in the proposed method is mostly due to the usage of CNN as a prior during optimization (which was discussed in supp. materials), which are exactly the contribution of DIP.)
> >
> > > The core of our framework lies in its ability to adapt to different tasks while maintaining a consistent structure.
> > The consistent architecture of $\alpha$-Net and RGBnet establishes a unified way of extracting the motion and appearance features from a video, across different tasks. These intermediate motion and appearance features are then utilized with task-specific adaptations that are reflected in the decomposition formulations. Hence, the unifying factor across different tasks is the consistency in the architecture used to extract motion and appearance cues from the videos across all tasks.
> >
> > (1) As you said, the intermediate features are utilized with task-specific adaptations with different formulations for different tasks. The formulation, prior loss functions, and the exact semantic meaning of the intermediate output from $\alpha$-Net and RGBnet, are completely different. The author(s) seems argue that "using the same network structure" is enough to regard it as a "unified framework". I agree that they are the same; but I would respectfully disagree for claim it as a "unified framework" if the only consistency between tasks is that they share a same network structure for feature extraction.
> >
> > (2) The same thing across the proposed method for different tasks is the usage a neural network to extract intermediate representations. Again, this idea itself has been discussed in many papers (e.g., the DIP work and its extensions).
> >
> > > Second, The VDP framework’s way of modeling motion and appearance in a video is unique to our work(utilizing
> > -Net and RGBnet). Specifically, to our knowledge, no prior test time optimization technique has utilized FlowRGBs to model the motion of the video, and it is unique to our methodology.
> >
> > I really agree with this point, and I think utilizing FlowRGBs as input is indeed the key from my understanding. Hence this paper would become far more interesting if it had dense discussions regarding this part (which unfortunately it does not).
> > The most interesting part here is that the while the input optical flow models the "relative" motion between two frames, the output of FlowRGBs (from the $\alpha$-net) becomes a per-frame $\alpha$ that represents some "absolute" status for static frames. I believe having an in-depth discussion from the FlowRGB input perspective is far more interesting than saying the proposed framework is "general".

---

> ### Comment · Reviewer_6Tks · 2023-11-18
> **Further comments (2/2)**
>
> > With regards to the mentioned test-time methods[2,3], they are not at all designed for the VOS task but more for edit propagation tasks. These methods require per-frame masks obtained by image segmentation baselines. So, we do not see any overlap of these techniques with our framework apart from edit propagation tasks.
>
> I was misunderstanding that your method also requires mask input for first frame. Thanks for the clarification.
>
> > We also would like to emphasize that maintaining spatio-temporal coherence is a big issue when it comes to tasks like dehazing and relighting, as can be seen from the qualitative results provided in the webpage(supplementary material). Our novelty also stems from the fact that our method can provide high-fidelity results with far superior spatio-temporal coherency.
>
> See comments above - I found it really hard to say the proposed method has "far superior spatio-temporal coherency" from the video results; For relighting task, it seems all methods except SDSD has similar level of temporal coherency.
> Also, it is more important to understand why the proposed method has temporal coherency than just say "we have good results". For example, is the good temporal coherency comes from the $\alpha$-map extracted from optical-flow inputs; or it is a directly outcome of applying optical-flow as a warping constraint as a loss function? The latter (warping constraints) is a common approach in video tasks, though.
>
> > In Table 2, we only compare with other test-time optimization baselines for a fair comparison as the setting of each methodology is exactly the same, i.e., no prior segmentation annotation is given to the model during training.
>
> Your setup is different from VOS as no initial mask is given. Thanks for clarification.
>
> > Time taken: We refer the reviewer to our appendix section C.
> I have read section C and still a little bit confused:
> - "We achieve a processing rate of 60 frames/min for dehazing and relighting and 40 frames/min for VOS with one salient object"
>
> Is this processing rate referring to optimization time or just inference time?
> - "We use 100 epochs for a 60-frame sequence in VOS and 60 epochs for dehazing and relighting"
>
> Let's make the answer simple - what exactly is the time for optimizing one video (60-frames, for example)?

---

> > ### Author Response · Authors · 2023-11-18
> > **Comprehensive Clarification (1/2)**
> >
> > We really appreciate the time taken by the reviewer to go through our response. We are very thankful to the reviewer for providing us with the critical feedback to improve our paper further. Below, we offer detailed clarifications to each point raised, aiming to address reviewer’s concerns comprehensively.
> >
> >
> > **Clarification on Novelty in Relighting Task:** We appreciate the reviewer's acknowledgment of the suitability of multiplicative decomposition for relighting tasks. Addressing the concern about the novelty of our approach, we reiterate our claim based on two key aspects:
> >
> > - *Novel Formulation in Decomposition:* The first aspect of our novelty claim is the unique decomposition formulation we came up with leveraging the gamma correction equation. This is a significant departure from the existing baseline methodology for video relighting task.
> >
> > - *State-of-the-Art Results and Validation:* The second aspect underpinning our claim is the high fidelity and state-of-the-art results achieved by our approach, as evidenced in both qualitative and quantitative evaluations. The qualitative results, which the reviewer commended as 'great', and the quantitative findings detailed in Table 1 of our paper, collectively highlight the efficacy of our method. To further substantiate our claim, we have conducted ablation studies to pinpoint the major contributors to these exceptional results.
> >
> > In light of these factors, we argue that our approach represents a substantial novel contribution to the field of relighting. It not only introduces a unique formulation for the task but also achieves state-of-the-art results, setting new benchmarks in the domain. Hence, we respectfully request the reviewer to reevaluate the novelty of our contribution, considering both the innovation in our method and the significant advancements it brings to relighting tasks.
> >
> >
> >
> > **Addressing Temporal Coherence Concerns:** We sincerely thank the reviewer for acknowledging the high quality of our relighting results. In response to the concerns about temporal coherence, we offer additional insights that underscore the distinctiveness of our method compared to ZeroDCE++ and StableLLVE.
> >
> > First, the ZeroDCE++ method is an image-based baseline. The basic flaw of applying an image-based baseline to a video task is it creates temporal aberrations in the form of jittering, which, in the case of the relighting task, comes in the form of noise. As the reviewer correctly highlighted, there is an existence of noise in the processed video results for Zerodce++. Hence, this method would be considered a temporally unstable method.
> >
> > As for StableLLVE, it indeed employs a dedicated module to enforce temporal consistency. However, this is achieved at the expense of the fidelity of processed sequences, a compromise clearly evidenced in both our qualitative and quantitative assessments, including video results and the data presented in Table 1. This fidelity trade-off is a crucial factor that our method adeptly navigates, balancing temporal coherence with high-quality output.
> >
> > Additionally, it is important to note that the metrics PSNR and SSIM, which are the standard metrics for the evaluation of such a task, are applied frame by frame on a video. Hence, these metrics do not account for temporal coherence in the results. Therefore, we also presented video results to assure reviewers that our method produces temporally stable results. Free from all temporal aberrations. The notable absence of noise in our method's output is indicative of superior temporal stability, a quality essential for high-fidelity video relighting.

---

> > > ### Author Response · Authors · 2023-11-18
> > > **Comprehensive Clarification (2/2)**
> > >
> > > **Discussion about FlowRGBs utilization as a novelty:** We agree with the reviewer’s point that we have not highlighted the importance of the utilization of FlowRGBs in the main paper to the fullest extent possible. We propose a way to rectify this in the following manner.
> > >
> > > First, we will mention the point that we are the first test-time optimization technique to exploit FlowRGB information as a proxy for understanding motion in a given video sequence as our major contribution in this work.  We have already performed extensive experimentation that can potentially highlight the contributions of FlowRGBs for all three tasks.
> > > - VOS: We derived a loss function(Flow similarity loss)  that is completely based on FlowRGB. In section D.4(appendix), it can be observed that neglecting this loss while optimization results in a significant drop. This observation solidifies the contribution of FlowRGB in this task.
> > >  - Relighting/Dehazing tasks: We performed extensive ablation studies for both of these tasks with a specific notion in mind to study the effects of FlowRGBs when performing these tasks. It can be observed from Table 4. (appendix), Fig. 4(appendix), and Section F (appendix) that there is a significant drop in both the quantitative and qualitative results when the flowRGB is not utilized as a proxy for motion cue of a video sequence highlighting its importance.
> > >
> > > We propose to have at least the tables from the appendix to the main paper. If ICLR allows extra pages, then the migration would be much easier. However, if this is not the case, we propose to create space in the main paper by moving the details about the dataset to the appendix and utilizing the space to highlight the contribution of the FlowRGBs.
> > > By making these amendments, we aim to align the paper more closely with the reviewer's expectations and demonstrate the true novelty and significance of our work. We hope these changes will make the paper more compelling and justify a reconsideration of its rating.
> > >
> > > **Addressing the Unified Approach Issue:**
> > > The term 'unified' has been used twice in our paper. However, following discussions with the reviewer, we are willing to eliminate this term completely, acknowledging that a unified framework is not the primary contribution of our work. To further clarify our contributions, we propose revising the initial claim in the paper.
> > >
> > > - *Original Claim:*
> > > "We propose a novel inference-time optimization framework that decomposes a video into multiple layers and can effectively perform tasks like video object segmentation, dehazing, and relighting."
> > >
> > >  - *Revised Claim:*
> > > "We propose a novel video decomposition prior that effectively exploits appearance and motion cues (in the form of FlowRGB) from a query video sequence to perform tasks like video object segmentation, dehazing, and relighting without requiring any task-specific data."
> > >
> > > This revision serves two primary purposes. First, it reframes the focus of our work around the exploitation of appearance and motion cues from videos, highlighting that this is key to performing UVOS, dehazing, and relighting tasks. It also suggests that the effective use of these cues can replace the need for task-specific data in UVOS, Relighting, and Dehazing tasks. Second, it shifts the focus away from the notion of a unified framework, more accurately reflecting the essence of our contribution.
> > >
> > > **Understanding the relighting results** We refer the reviewer to section F of the appendix, where we perform extensive ablation studies to figure out the components contributing to our results. We hope these clarifications alleviate the reviewer's concern.
> > >
> > >
> > > **Time taken** example: VOS task for one video with 60 frames: the time required to perform inference time optimization for vos task on an average is 40 frames/min. Hence, it would take for a video with 60 frames $\frac{60}{40}$mins or 1min:30 secs. Hopefully, this clarifies the reviewer’s question.

---

> > > ### Comment · Reviewer_6Tks · 2023-11-20
> > > **Updating score to 6**
> > >
> > > Thanks for the author(s) response and discussion. After reading the latest response I decided to update my score to 6 based on the following reason -
> > >
> > > (1) The author(s) has decided to eliminate the "unified framework" to make the paper positioned in a more accurate way in the revised version.
> > >
> > > (2) The optical-flow input coupled with DIP-style prior for test-time optimization is new and interesting, and the author(s) will discuss more on this.
> > >
> > > (3) I still have concerns regarding the novelty of relighting task formulation and the evaluation on temporal coherence; yet it is a minor issue that could be somehow addressed.
> > >
> > > (4) My other concerns have been clarified and addressed.
> > >
> > > Given the current status I would update the score to 6; I would say it hard to further raise the score because this paper (even accepted) would require a (somehow major) revision of the overall flow and texts.

---

### Official Review · Reviewer_XPSu · 2023-10-30

**Soundness:** 3 good
**Presentation:** 3 good
**Contribution:** 2 fair
**Rating:** 6
**Confidence:** 4

**Summary:**

This paper introduces a new approach called the Video Decomposition Prior (VDP) framework. Unlike conventional methods, VDP leverages the motion and appearance of the input video, decomposing it into multiple RGB layers with associated opacity levels. These layers are then manipulated individually to achieve the desired results, addressing tasks like video object segmentation, dehazing, and relighting. The paper also introduces a logarithmic video decomposition formulation for relighting tasks. The approach is evaluated on standard video datasets, including DAVIS, REVIDE, and SDSD, demonstrating qualitative results on a diverse range of internet videos.

**Strengths:**

+ The proposed video decomposition prior leverages the motion and appearance of the input video, decomposing it into multiple RGB layers with associated opacity levels.
+ The proposed VDP is employed in different video-based tasks, including video object segmentation, video dehazing and video relighting.

**Weaknesses:**

- In the paper, the limitations of the VDP is not discussed in the paper, and all the results are good cases.
- In the introduction of Flow similarity loss, the VGG embeddings of masked flow-RGB and those of other layers are used to calculate the cosine similarity. It is unclear how to generate VGG embeddings from masked feature maps for cosine similarity calculation.
- In equation (12), the behavior of the reconstruction layer loss resembles that of the L1 loss in the reconstruction loss. To validate the rationale behind the design, it is essential for the paper to elucidate the distinctions between them and elucidate the impact of the reconstruction layer loss.

**Questions:**

Please refer to the questions in the weakness.

---

> ### Author Response · Authors · 2023-11-12
>
> We thank the reviewer XPSu for your thoughtful and detailed review of our paper on the Video Decomposition Prior (VDP) framework. We appreciate your recognition of our novel approach and its applications in video processing tasks. We have addressed each of your concerns below, aiming to clarify the aspects you highlighted. We believe these clarifications and additional information will solidify the strength of our paper and align it more closely with the acceptance criteria for ICLR.
>
> **1. Limitation of VDP not discussed:**
> We acknowledge the importance of discussing the limitations of our methodology. To address this, we have added a comprehensive discussion on the limitations of VDP in Section H of our appendix (part of the supplementary material). Due to space constraints, this section was not included in the main paper, but we are prepared to move it to the main body if ICLR allows an additional page in the final submission. This inclusion will provide a more balanced view of our methodology and its applicability.
>
> **2. Flow Similarity Loss:**
> We appreciate your query regarding the formulation of our Flow Similarity Loss. This loss is crucial for differentiating the motion of the background from the foreground. The process involves three steps:
>
> - **Step 1:** We take the FlowRGB of the frame at time step $t$ (the process of obtaining FlowRGBs is detailed in Section K of the appendix included in supplementary material) and perform element-wise multiplication with both the foreground mask and the background mask, i.e.,
>   - $FlowRGB_{Fg} = FlowRGB^t \odot M^t$
>   - $FlowRGB_{Bg} = FlowRGB^t \odot (1-M)^t$
>
> - **Step 2:** We use these obtained $FlowRGB_{Fg}$ and $FlowRGB_{Bg}$ and find their corresponding VGG embeddings. This is done by passing them as input to the VGG encoder (pretrained on ImageNet dataset and denoted as $\phi(.)$) and taking the output of the last layer of the VGG encoder as the embedding vector. Mathematically, we can write these embedding vectors for the foreground and background layer as,
>   - $embed_{Fg} = \phi( FlowRGB_{Fg})$
>   - $embed_{Bg} = \phi( FlowRGB_{Bg})$
>
> - **Step 3:** In this final step, we calculate the loss value by taking a cosine value between these embeddings of the foreground and background layer.
>   - $L_{FlowSim} = \frac{embed_{Fg} . embed_{Bg}}{|embed_{Fg}| |embed_{Bg}|}$
>
> We minimize the cosine similarity between the VGG embeddings of the background layer and the foreground layer, which translates to the learning of a mask that separates the motion of the background and foreground. We hope this detailed explanation resolves any ambiguities about flow similarity loss.
>
> **3. Ablation Study Involving Reconstruction Layer Loss:**
> We understand the concerns of the reviewer about the understanding the impact of reconstruction layer loss. We would refer the reviewer to section D.4 of the appendix(part of supplementary) where we perform extensive experiments on the DAVIS-16 dataset to study the impact of each loss term on the final segmentation results. We hope our clarifications have addressed the concerns of the reviewer.
>
> We hope that our responses provide the necessary clarifications and insights into our methodology, reinforcing the potential impact and novelty of our work in the field of video processing. We are optimistic that these clarifications justify a rating upgrade from the current rating of "6".

---

### Official Review · Reviewer_8QGm · 2023-10-31

**Soundness:** 3 good
**Presentation:** 3 good
**Contribution:** 2 fair
**Rating:** 6
**Confidence:** 4

**Summary:**

This paper introduces the Video Decomposition Prior (VDP) framework, a new approach for video editing tasks, including object segmentation, dehazing, and relighting. VDP decomposes videos into multiple layers and optimizes parameters without explicit training. The proposed logarithmic video decomposition enhances video relighting, resulting in state-of-the-art performance in downstream tasks: unsupervised video object segmentation, dehazing, and relighting.

**Strengths:**

1. Sound approach: The VDP framework presents an innovative approach to video editing, offering practicality and cost-effectiveness by not relying on extensive datasets or ground truth annotations. The ability to optimize parameters using the test sequence itself distinguishes VDP from traditional deep learning methods, which often require extensive training data.

2. State-of-the-Art Performance: VDP demonstrates top-tier performance in key downstream tasks, including unsupervised video object segmentation, dehazing, and relighting.

3. Good writing and representation.

**Weaknesses:**

I am generally positive about this paper. my main concern lies in the lack of comprehensive comparison: The paper does not provide a comprehensive comparison with other video editing techniques, making it difficult to assess the VDP framework's performance against other state-of-the-art methods. Furthermore, video editing is a broad field, including tasks such as adding or removing objects. It seems that the proposed method may not be suitable for handling these scenarios. Therefore, I suggest that the authors consider refining the title and corresponding claims.

**Questions:**

Please see the weakness

---

> ### Author Response · Authors · 2023-11-12
>
> We thank the reviewer 8QGm for your thorough review and constructive feedback on our submission, "Video Decomposition Prior (VDP)." We appreciate your recognition of VDP's innovative approach and its state-of-the-art performance in unsupervised video object segmentation, dehazing, and relighting. Your insights are invaluable, and we have addressed your concerns below, hoping to strengthen our paper and align it more closely with your expectations.
>
> **Addressing the Comprehensive Comparison Concern:**
> You rightly pointed out the necessity for a comprehensive comparison with existing video editing techniques. To address this, we have included additional comparative analysis in Section G of our appendix(supplementary material). This section details a unique proxy task developed specifically to evaluate VDP against contemporary baselines, considering the absence of a standardized comparison framework in video editing. The results of this comparative analysis are presented in Table 6 of the appendix. Additionally, we have prepared a supplementary webpage featuring video results and qualitative comparisons with these baselines, which should provide a more tangible understanding of VDP's capabilities in relation to existing methods.
>
> **Refining the Title and Scope:**
> In response to your suggestion, we agree that refining the title and scope of our paper could more accurately represent the specific editing tasks that VDP excels in. We propose the revised title, "Video Decomposition Prior and its Applications in Segmentation, Dehazing, and Relighting." This modification narrows down the scope and directly reflects the core strengths of our methodology. We are open to further suggestions from the reviewer regarding the title and scope adjustments.
>
> We believe these updates and clarifications directly address your concerns and enhance the paper's relevance and contribution to the field. We hope these modifications and clarifications resonate with your assessment criteria and upgrade the paper's rating from a ‘6'.

---

### Official Review · Reviewer_u6Cn · 2023-10-31

**Soundness:** 2 fair
**Presentation:** 2 fair
**Contribution:** 3 good
**Rating:** 5
**Confidence:** 4

**Summary:**

This paper attempts to propose a general framework for video editing. It starts by predicting the video into several individual layers and then utilizes task-specific knowledge to reconstruct the video. A reconstruction loss and a warping loss are utilized to train the network. This is an inference-time optimization framework that does not rely on external training tools. However, the framework is rather standard, and the choice of a few loss functions for the decomposition tasks is straightforward. Overall, the novelty is limited. I am inclined to reject this article. Please check the details in other sections.

**Strengths:**

The motivation to propose such a general framework is good.

**Weaknesses:**

-: The reconstruction loss and warp loss used are very common loss functions and do not offer any novelty.

-: The method is still not sufficiently general. For example, for the task of video segmentation, additional loss functions need to be designed as constraints. The three task-specific losses mentioned in Section 3.2 are the only ones provided. From this perspective, it is difficult to see what this framework proposed in the paper brings to this community.

-: The comparison results of the experiments are not that fair. For instance, in Table 3, the comparison is made with the latest algorithm, CG-IDN (a 2021 algorithm). There are many dehazing algorithms that could be compared, including single-image hazing algorithms with stability processing (references [1][2]).

-: Additionally, Table 2 lacks many baselines. Looking at the official website of DAVIS2016, the best baseline achieves an IOU score of over 82. Why wasn't this paper compared against that?

-: Sometimes it is necessary to introduce additional prior knowledge, such as for the task of dehazing, where the effectiveness is actually limited.

[1] Learning Blind Video Temporal Consistency

[2] Blind Video Temporal Consistency via Deep Video Prior

**Questions:**

See weakness part.

---

> ### Author Response · Authors · 2023-11-11
>
> We thank the reviewer for taking their time out of a busy schedule and evaluating our submission. We are glad that reviewer u6Cn found our paper well-motivated. Before we address the specific weaknesses and questions of the reviewer, **we highly recommend reviewer u6Cn to please go through the webpage of video results**, which was attached in the supplementary file.
>
>
> **Novelty issues:**
>
> We understand the concern raised by the reviewer over the novelty of the paper. Here is a more structured view of our model. The core of our framework for any task at hand has three components to it: prior knowledge(decomposition formulation), motion modeling (via $\alpha$-Net), and appearance modeling (via RGBnet). The framework’s way of modeling motion and appearance in a video is unique to our work. Specifically, to our knowledge, no prior test time optimization technique has utilized FlowRGBs to model the motion of the video.
>
> Second, the utilization of prior knowledge(from gamma correction) to derive a novel decomposition formulation for the relighting task is a unique contribution of our work.
> Lastly, we are able to achieve state-of-the-art temporally coherent results utilizing our framework for the task of dehazing and relighting. We would urge the reviewer to please check out the difference in the qualitative video results of our methodology and baselines(We have provided these in an HTML webpage as a part of our supplementary material). To the reviewer’s point, our work derives the novelty from the above-mentioned points as opposed to the reconstruction or warping loss. We hope our clarification will alleviate the novelty concerns of the reviewer.
>
>
> **What is the consistency across tasks:**
>
> We thank the reviewer for pointing out the difficulty in understanding the consistency between the tasks. Hopefully, our clarification about it will alleviate this difficulty.
>
> The core of our framework lies in its ability to adapt to different tasks while maintaining a consistent structure. Basically, each task consists of three integral components: prior knowledge (decomposition formulation), motion modeling (via $\alpha$-Net), and appearance modeling (via RGBnet). The **consistent architecture** of $\alpha$-Net and RGBnet establishes a **unified** way of extracting the motion and appearance features from a video across different tasks. These intermediate motion and appearance features are then utilized with task-specific adaptations that are reflected in the decomposition formulations. Hence, the unifying factor across different tasks is the consistency in the architecture used to extract motion and appearance cues from the videos across all tasks.
>
> We would put more effort into revamping the introduction section of our manuscript to emphasize this point of consistency our framework's architecture brings across this variety of tasks.
>
>
> **Baseline comparison for UVOS Task:**
>
> We thank the reviewer for raising this point. In Table 2, we only compare with other test-time optimization baselines for a fair comparison as the setting of each methodology is exactly the same, i.e., no prior segmentation annotation is given to the model during training. We provide a comprehensive evaluation of the task UVOS task in Table 5 in the appendix because of the limited availability of space in the main paper. We would add the results to the main paper if ICLR allows one extra page in the final camera-ready paper.
>
>
> **Additional Experiments:**
>
> We provide more comparisons with the baseline suggested by the reviewers for the dehazing task. We utilize the MSBDN as an image dehazing baseline. Please find the findings below.
> | Metrics | Blind[1] | DVP[2] | Ours |
> |----------|----------|----------|----------|
> | PSNR|22.98| 23.05 | 24.83|
> |SSIM| 0.877 | 0.882 | 0.916|
>
>
> **Additional Prior knowledge for the dehazing task:**
>
> We agree with the reviewer that additional knowledge about the scene can improve the efficacy of our methodology. However, it is very difficult to acquire hazy video and clean video pairs in a real-world setting. Hence, we formulated a test time optimization technique that does not rely on task-specific data for performing the dehazing.

---

> > ### Author Response · Authors · 2023-11-19
> >
> > We sincerely thank the reviewer for dedicating their time and expertise to assess our paper. To the best of our knowledge, we believe that our rebuttal has effectively addressed the concerns raised, and we are grateful for the reviewer's thoughtful feedback.
> >
> > As the authors-reviewers discussion period enters its final days, we wish to reiterate our commitment to providing any additional clarification that may be required. Should there be any concerns that necessitate a more comprehensive explanation, we are more than willing to accommodate such requests promptly.
> >
> > Sincerely,
> >
> > Authors

---

> > ### Comment · Reviewer_u6Cn · 2023-11-21
> > **Further comments**
> >
> > Thank the authors for the detailed feedback. The rebuttal of the authors addresses most of my concerns, and I have some further questions regarding the feedback.
> >
> > - The novelty of FlowRGB: I am not quite clear about the argument "no prior test time optimization technique has utilized FlowRGBs to model the motion of the video". For example, it seems that [1] uses both flow and RGB for test-time optimization.
> >
> > - Are you going to remove the description about "Unified Approach". To me, your applications on three different tasks are still quite task-specific. It seems that you are going to revise the description in your feedback to other reviewers.
> >
> > - For VOS comparison, it is acceptable that you only compare with other test-time optimization approach. However, doesn't this go against your original intention? You should conduct experiments on some tasks that achieve better performance on test-time optimization approach to support your argument.
> >
> > - I appreciate the novelty on the relighting task and believe the your performance is better than other baselines.
> >
> > - Thanks for the results of experiments on dehazing tasks. It helps support your argument.
> >
> > Thanks.
> >
> > [1] Deformable Sprites for Unsupervised Video Decomposition

---

> > > ### Author Response · Authors · 2023-11-21
> > >
> > > We would like to express our gratitude for taking the time to review our paper and for providing us with valuable feedback. We are pleased to hear that our previous rebuttal addressed most of your concerns, and we are committed to addressing any remaining questions you may have.
> > >
> > > **The novelty of FlowRGB:** It is important to distinguish between the RGB frame and the FlowRGB representation in our paper. The RGB frame represents the standard 3-channel colored frame, while in our work, FlowRGB is defined as the Image/RGB representation of optical flow offsets (for additional details, please refer to Appendix Section K). These FlowRGB representations serve as feature maps input to the $\alpha$-net. While it is true that [1] employs optical flow and RGB data for test-time optimization, the crucial differentiation lies in the fact that they do not leverage FlowRGB for making predictions. In contrast, our approach uniquely utilizes image representations of optical flow offsets as feature maps in our architecture. This novel use of FlowRGB as a proxy for motion cues within video sequences constitutes a fundamental contribution of our work, resulting in a significant enhancement in our model's performance.
> > >
> > >
> > > **Unified approach resolution:** Yes, the reviewer is absolutely right. As we previously communicated with reviewer 6Tks, we will shift focus from the term `unified approach' and would be committed to providing a comprehensive and in-depth discussion about FlowRGBs in our revised manuscript. Our primary focus will be on highlighting the novelty aspect of our work, where we are the first to employ FlowRGBs as feature maps in the context of test-time optimization techniques. Moreover, we have already conducted rigorous evaluations to demonstrate the significant performance improvement achieved by utilizing these FlowRGB representations as feature maps, as illustrated in Figure 4 and Table 4 in the appendix.
> > >
> > >
> > > **VOS:** We are happy to hear that the reviewer finds our evaluation for the VOS task with other test-time optimization techniques acceptable. Additionally, we would like to mention that DoubleDIP technique is a test-time optimization technique that exists for video dehazing task. We compared DoubleDIP against our method and reported both qualitative(Fig 7(main paper), video results in supp.) and quantitative(Table 3(main paper)) results . However, for the relighting task we were not able to find such baselines. We want to thank the reviewer for pointing this out and we would make sure to highlight this baseline to make it easier for the reader to make the distinction.
> > >
> > >
> > > Lastly, we would like to express our appreciation for your recognition of the novelty and superior performance of our method in the relighting task. Your feedback has been valuable in refining our paper, and we hope that the clarifications provided in this response address your concerns. We kindly request that you reconsider your rating for our submission.
> > >
> > >
> > > Sincerely,
> > >
> > > Authors

---

> > > > ### Comment · Reviewer_u6Cn · 2023-11-22
> > > >
> > > > Thank you for your time and explanations. I believe your explanations solve most of my concerns. In summary, I think the major problem of this problem is the argument about "Unified". As Reviewer 6Tks says, this paper requires substantial revision to emphasize its real contribution and avoid confused arguments. Now, I hold a positive attitude toward this paper, and my major concern is: can the current paper (at current status) be accepted? If the revision can be promised and all reviewers feel OK about it. I think this paper can be accepted. In summary, rejecting or accepting this paper are both OK for me.

---

### Meta-Review · Area_Chair_gfws · 2023-12-04

**Metareview:**

The submission proposes a framework that decomposes a video into multiple multiple RGB layers and associated opacity levels for various video editing tasks. It uses two neural network modules, RGB-net and α-net, to predict RGB layers and opacity/transmission layers for each video frame. These layers are then composited to reconstruct the input video or achieve the desired effects during inference time optimization. The paper conducted experiments on video relighting, dehazing, and VOS tasks and achieves better results compared to some existing baselines.

Pros:
* well-motivated and sound approach
* good performance on a variety of tasks
* clear writing

Cons:
* novelty is somehow lacking as the the test-time optimization approach is standard
* the manuscript needs major revision for the camera ready

**Justification For Why Not Higher Score:**

While reviewers were overall positive about the paper, there were still concerns about novelty and whether the authors can finish a major revision of the manuscript to fully address comments by reviewers.

**Justification For Why Not Lower Score:**

All four reviewers were positive about the submission. Although giving a rating of 5,  reviewer u6Cn was mostly positive about the submission and the concern was whether the authors can make all the promised changes in the final version.

---

### Decision · Program_Chairs · 2024-01-16

Accept (poster)